# Engineered Multivalent Nanobodies Efficiently Neutralize SARS-CoV-2 Omicron Subvariants BA.1, BA.4/5, XBB.1 and BQ.1.1

**DOI:** 10.3390/vaccines12040417

**Published:** 2024-04-15

**Authors:** Jiali Wang, Bingjie Shi, Hanyi Chen, Mengyuan Yu, Peipei Wang, Zhaohui Qian, Keping Hu, Jianxun Wang

**Affiliations:** 1School of Life Sciences, Beijing University of Chinese Medicine, Beijing 100029, China; 2NHC Key Laboratory of Systems Biology of Pathogens, Institute of Pathogen Biology, Chinese Academy of Medical Sciences and Peking Union Medical College, Beijing 100730, China; 3The Institute of Medicinal Plant Development, Chinese Academy of Medical Sciences and Peking Union Medical College, Beijing 100730, China; 4Andes Antibody Technology Hengshui LL Company, Hengshui 053000, China; 5Shenzhen Research Institute, Beijing University of Chinese Medicine, Shenzhen 518118, China

**Keywords:** SARS-CoV-2, Omicron subvariants, phage display library, receptor-binding domain (RBD), multivalent nanobodies, neutralizing activity

## Abstract

Most available neutralizing antibodies are ineffective against highly mutated SARS-CoV-2 Omicron subvariants. Therefore, it is crucial to develop potent and broad-spectrum alternatives to effectively manage Omicron subvariants. Here, we constructed a high-diversity nanobody phage display library and identified nine nanobodies specific to the SARS-CoV-2 receptor-binding domain (RBD). Five of them exhibited cross-neutralization activity against the SARS-CoV-2 wild-type (WT) strain and the Omicron subvariants BA.1 and BA.4/5, and one nanobody demonstrated marked efficacy even against the Omicron subvariants BQ.1.1 and XBB.1. To enhance the therapeutic potential, we engineered a panel of multivalent nanobodies with increased neutralizing potency and breadth. The most potent multivalent nanobody, B13-B13-B13, cross-neutralized all tested pseudoviruses, with a geometric mean of the 50% inhibitory concentration (GM IC_50_) value of 20.83 ng/mL. An analysis of the mechanism underlying the enhancement of neutralization breadth by representative multivalent nanobodies demonstrated that the strategic engineering approach of combining two or three nanobodies into a multivalent molecule could improve the affinity between a single nanobody and spike, and could enhance tolerance toward escape mutations such as R346T and N460K. Our engineered multivalent nanobodies may be promising drug candidates for treating and preventing infection with Omicron subvariants and even future variants.

## 1. Introduction

A beta coronavirus known as severe acute respiratory syndrome coronavirus 2 (SARS-CoV-2) is to blame for the unprecedented coronavirus disease 2019 (COVID-19) pandemic [1,2]. As of 21 October 2023, the global tally of confirmed COVID-19 cases surpassed 771 million, with a staggering death toll of 6.9 million individuals (https://covid19.who.int/ (accessed on 21 October 2023)). Moreover, this unprecedented pandemic has profoundly disrupted economies and strained healthcare systems worldwide. The scientific community has been diligently striving to develop effective treatments and vaccines against SARS-CoV-2. Therapeutic neutralizing antibodies represent a critical class of antiviral treatments that have been rapidly developed and deployed within a short period for the prophylaxis and treatment of SARS-CoV-2 [3]. However, the ongoing adaptive evolution of SARS-CoV-2 has resulted in the emergence of several variants of concern (VOCs), which pose a significant challenge to the effectiveness of antibody treatments. This is particularly true for Omicron and its descendant subvariants, as they have not only impaired the efficacy of most neutralizing antibody treatments, but also undermined the effectiveness of existing vaccines [4,5,6,7].

The original Omicron variant (B.1.1.529) was first detected in South Africa toward the end of 2021 and quickly gained worldwide attention as it rapidly spread across different countries and continents [8]. Compared to other VOCs, Omicron exhibits higher transmissibility and stronger immune evasion due to the presence of numerous mutations within the spike (S) protein [4,9,10]. Over time, Omicron has expanded into multiple subvariants, such as BA.1, BA.2, BA.2.75, BA.4 and BA.5 (hereafter BA.4/5, as these two share an identical spike sequence), BF.7, BQ.1.1 and XBB; these subvariants have further exacerbated public health concerns. In particular, BA.4/5 subvariants can further evade acquired immunity developed from previous infections with BA.1 or BA.2 subvariants, resulting in a surge of breakthrough infections in numerous regions globally [11,12]. Subsequently, novel subvariants BQ.1.1 and XBB.1 have emerged and supplanted BA.5 as the prevailing variants due to their stronger ability to evade antibodies [13,14]. Notably, all clinically authorized therapeutic antibodies were rendered inactive against subvariants BQ.1.1 and XBB.1 [13]. Moreover, a BA.5 bivalent booster was found to fail to elicit a robust immune response against BQ.1.1 and XBB.1, further highlighting the remarkable antibody evasion properties of these two subvariants [15]. Therefore, it is imperative to develop potent and broad-spectrum neutralizing antibodies for the effective management of Omicron subvariants and proactive preparedness against future emerging variants.

The spike (S) glycoprotein of SARS-CoV-2, which is composed of the S1 and S2 subunits, plays a crucial role in infecting host cells [16,17,18]. The S1 subunit can be divided into an N-terminal domain (NTD) and a receptor-binding domain (RBD) [18]. Within the RBD, there is a receptor-binding motif (RBM) responsible for recognizing and binding to a specific host cell receptor known as angiotensin-converting enzyme 2 (ACE2) [19,20]. Once bound to ACE2, the S2 subunit facilitates viral fusion with host cell membranes [21,22]. Based on this infection process, the spike glycoprotein is considered an effective target for neutralizing antibody therapy. The current classification of neutralizing antibodies against the spike glycoprotein includes three distinct categories: those that target the NTD, those that target the RBD, and those that target the S2 subunit [23]. The majority of the highly potent neutralizing antibodies developed thus far have predominantly focused on targeting the RBD [23,24,25]. Their RBD epitopes are grouped into seven core communities, which are located on the cryptic inner face, the solvent-exposed outer face, and the top RBM face of the RBD [26]. Neutralizing antibodies that specifically bind to the RBM can directly inhibit the interaction between ACE2 and the spike glycoprotein, thereby effectively blocking viral entry [27,28]. Other neutralizing antibodies that bind to the non-RBM region prevent viral entry by sterically inhibiting ACE2 binding, destabilizing the spike trimer, or blocking spike-mediated syncytia formation [29,30,31]. Notably, the majority of spike glycoprotein mutations in Omicron are concentrated in the RBD region, significantly impairing the potency of most RBD-targeting neutralizing antibodies [32]. Recently, several broadly neutralizing antibodies have been reported to demonstrate potent efficacy in neutralizing BA.4/5, BQ, and XBB subvariants by binding to highly conserved RBD epitopes [33,34,35,36]. These studies have demonstrated that the RBD still contains protective and conserved epitopes, which can serve as viable targets for the development of potent and broad-spectrum neutralizing antibodies.

Variable heavy domains of heavy chains (VHHs), also known as nanobodies, are derived from heavy-chain-only antibodies present in Camelidae family members, including alpacas, llamas, and camels [37]. Nanobodies, which are the smallest naturally occurring proteins known to possess the ability to bind to antigens, have molecular weights ranging from 12 to 15 kDa. They are commonly generated through phage display using naïve, immune, or synthetic libraries [38]. Compared to traditional antibodies, nanobodies offer numerous benefits, including high thermostability and solubility, low immunogenicity, ease of rapid production, capacity to bind cryptic epitopes, and potential for engineering into multivalent forms with enhanced functionalities [39,40]. Furthermore, nanobodies can be effectively delivered via inhalation due to their small size and excellent stability, which makes them highly suitable for the prevention and treatment of respiratory ailments [41]. Because of these advantageous characteristics, nanobodies are regarded as a promising option for next-generation treatments against SARS-CoV-2 infection. In recent years, significant advancements have been made in the development of nanobodies for combating COVID-19 [42,43]. However, the repertoire of nanobodies exhibiting exceptional potency and broad-spectrum activity remains significantly limited.

Here, we have constructed a highly diverse nanobody phage display library and identified nine nanobodies that target multiple distinct epitopes on the RBD. Five of them cross-neutralized the SARS-CoV-2 wild-type (WT) strain and Omicron subvariants BA.1 and BA.4/5, with one specific nanobody (A14) exhibiting effective neutralization even against subvariants BQ.1.1 and XBB.1. Furthermore, we selected four neutralizing nanobodies that target three nonoverlapping epitopes to engineer various multivalent nanobodies, including two homodimers, five heterodimers, one homotrimer, and one heterotrimer. Surprisingly, most of the multivalent nanobodies showed enhanced neutralizing capacity and inhibited escape for the subvariants BQ.1.1 and XBB.1. The most potent B13-B13-B13 homotrimer could efficiently neutralize all tested Omicron subvariants at low concentrations. Finally, we clarified the underlying mechanisms by which the multivalent nanobodies increased their neutralization breadth. In summary, the rational engineering of multivalent nanobodies provides a rapid approach to generating highly potent viral neutralizers for effectively controlling viral escape mutants

## 2. Materials and Methods

### 2.1. Cell Lines, Plasmids, and Recombinant Proteins

Both HEK293T and HEK293-hACE2 cells were cultured in Dulbecco’s modified Eagle’s medium (DMEM, Gibco, Carlsbad, CA, USA) supplemented with 10% fetal bovine serum (FBS, TransGen Biotech, Beijing, China). The culture of HEK293-hACE2 cells needed the addition of an extra concentration of 200 μg/mL G418 (Beyotime, Shanghai, China). Plasmids encoding the spike glycoprotein of SARS-CoV-2 WT, BA.1, BA.4/5, BQ.1.1 or XBB.1, along with the lentiviral packaging plasmid psPAX2 and the reporter plasmid pLenti-GFP expressing GFP and luciferase, were generously provided by Zhaohui Qian (Chinese Academy of Medical Sciences and Peking Union Medical College, Beijing, China). Plasmids encoding the spike glycoprotein of Omicron BA.4/5 were subjected to site-directed mutagenesis, as previously described, to introduce single-point mutations [44]. Sino Biological provided all the proteins used, including SARS-CoV-2 wild-type (WT) RBD with an Fc tag, Omicron BA.1 RBD with an mFc tag, Omicron BQ.1.1 spike trimer with a His tag, Omicron XBB.1 spike trimer with a His tag, and biotinylated hACE2 with a His tag.

### 2.2. Construction of the Camelid Nanobody Phage Display Library

Andes Antibody Technology Hengshui LL Company immunized four alpacas using recombinant S1 and RBD proteins from the wild-type strain of SARS-CoV-2. Total RNA was extracted from the peripheral blood mononuclear cells (PBMCs) of these alpacas and then reverse-transcribed into cDNA using gene-specific primers, oligo (dT), or random hexamers (Appendix A) [45]. The VHH genes were amplified using specific oligonucleotides through nested PCRs (Appendix A), followed by the insertion of the purified VHH fragments into the phagemid vector pHEN1 (HedgehogBio) via the SfiI/NotI restriction sites. These recombinant plasmids were introduced into *E. coli* TG1 cells (Renyu Biotechnology, Chengdu, China) via electroporation at 2.5 kV for a duration of 5 ms. After being revived in the shaker, the transformants were evenly spread onto six 15 cm-diameter 2YT agar plates supplemented with 100 μg/mL ampicillin (Lablead, Beijing, China). Subsequently, the colonies were harvested and transferred to 2YT medium after overnight incubation at 37 °C. The plating of serial dilution aliquots was used to calculate the size of the VHH library. Twenty-four colonies were randomly picked from serially diluted plates and subjected to colony PCR assay for the validation of VHH inserts. For the preparation of the phage library, the *E. coli* library was inoculated to 250 mL of 2YT medium supplemented with 2% glucose and 100 μg/mL ampicillin (2YT-GA) and grown until the OD600 reached 0.5. Subsequently, the bacterial cultures were infected with M13KO7 helper phages (New England Biolabs, Beverly, MA, USA) and incubated overnight. Finally, a nanobody phage display library was obtained using the PEG/NaCl precipitation method.

### 2.3. Phage Biopanning

Briefly, three rounds of biopanning were conducted to obtain nanobodies that target the RBD with high affinity. The quantity of WT RBD protein coated in the microtitre well gradually decreased after each round of panning. After three PBST rinses, the RBD well was blocked using 300 μL of blocking solution (2% BSA in PBST) and incubated at 37 °C for one hour. Concurrently, the nanobody phage display library (~10^11^ phage particles, approximately 10,000 × library size) was added to the control well without the RBD antigen for preincubation to eliminate the unspecific phages. After three PBST rinses, the preincubated nanobody phage display library was added to the RBD well and incubated at 37 °C for two hours. Next, the well was washed 20 times with PBST, and the bound phages were released by incubating them in a 10 μg/mL trypsin solution (Solarbio, Beijing, China). After infecting exponentially growing *E. coli* TG1, the released phage particles were amplified and utilized for the subsequent round of biopanning.

### 2.4. Phage ELISA

A total of 106 individual clones obtained from the third round of biopanning were measured using a phage ELISA. Each clone was inoculated into 1 mL of 2YT-GA and grown until the OD600 reached 0.5. After being infected with the M13KO7 helper phage, the bacterial cultures were incubated overnight at 30 °C. Following centrifugation, 500 μL of phage supernatants was collected for subsequent phage ELISA.

The wells of a 96-well microplate (Costar, Corning, NY, USA) were coated with 50 ng of WT RBD, while the negative control consisted of 5% BSA. Following an overnight incubation at 4 °C, a blocking solution of 300 μL/well was added to the microplate for blocking. Subsequently, 100 μL of each phage supernatant was added to the wells containing RBD and BSA. After incubation at 37 °C for one hour, the microplate was washed three times with PBST to remove nonbinding phages. The bound phages were detected by incubating them with HRP-conjugated anti-M13 antibody (1:10,000, Sino Biological) at 37 °C for one hour. Then, TMB substrate and stop solution (Solarbio) were sequentially added. The absorbance at 450 nm was measured using a SpectraMax i3x plate reader (Molecular Devices, San Jose, CA, USA). The RBD and BSA well reads were recorded, and the binding ratio was calculated.

### 2.5. Reformatting, Expression, and Purification of Nanobodies

Monomeric nanobody coding sequences were directly cloned into the pET-22b (+) vector with a C-terminal His-tag (HedgehogBio) using Gibson cloning. For multivalent nanobodies, two or three monomeric nanobody coding sequences were first connected head-to-tail through (Gly_4_Ser)_4_ linkers via overlap extension PCR and subsequently cloned into the pET-22b (+) vector. All recombinant plasmids were transformed into competent BL21(DE3) cells (Solarbio). Single colonies were inoculated into 5 mL of LB medium containing 100 μg/mL ampicillin (LB-A). After overnight incubation at 37 °C, the precultures were diluted at a ratio of 1:100 with 50 mL of LB-A and grown until the OD600 ranged from 0.5 to 1.0. Subsequently, the bacterial cultures were treated with 0.5 mM IPTG (Solarbio) and incubated at 30 °C for a duration of 4 h to induce the production of recombinant nanobodies. After centrifugation, the bacterial cultures were resuspended in lysis buffer and then subjected to ultrasonication on ice. The clarified bacterial lysates were collected by centrifugation at 12,000× *g* for 20 min. To obtain the purified nanobodies, the bacterial lysates were loaded onto Ni Sepharose 6 Fast Flow (GE Healthcare, Marlborough, MA, USA), followed by gradient elution using prechilled elution buffers (pH 7.4) containing 100–300 mM imidazole (Solarbio). The 3 kDa MWCO centrifugal filter units (Pall, New York, NY, USA) were used to concentrate the eluted nanobodies and exchange their buffer into 1×PBS. The final concentrations of all nanobodies were measured using the BCA method, and their purity was assessed by resolving them on 12% SDS–PAGE gels.

### 2.6. ELISA

The 96-well microplates were supplemented with 50 ng/well of WT RBD, BA.1 RBD, BQ.1.1 S-trimer, or XBB.1 S-trimer proteins and incubated overnight at 4 °C. After three PBST rinses, 300 μL/well of blocking solution was added to the microplates for blocking. The nanobodies and ACE2 were diluted in a 5-fold gradient with 0.5% BSA/PBST and then added to the microplates for incubation at 37 °C for one hour. Subsequent experiments were conducted as described above using HRP-conjugated anti-His antibody (1:5000, Proteintech, Chicago, IL, USA), TMB substrate, stop solution, and a plate reader. The experiments were conducted in duplicate, and the mean of the OD_450_ readings was used to calculate the EC_50_ value.

### 2.7. Surface Plasmon Resonance (SPR)

A Biacore T200 instrument (Cytiva Life Sciences, Marlborough, MA, USA) was used to perform SPR measurements at 25 °C. SARS-CoV-2 WT RBD and BA.1 RBD were immobilized on a CM5 Series S sensor chip (Cytiva Life Sciences) with approximately 300 response units (RUs) following dilution in sodium acetate buffer (pH 5.5). Fivefold serial dilutions of purified nanobodies were used as the mobile phase and flowed through the sensor chip at a flow rate of 30 μL/min for 180 s, followed by a dissociation flow that lasted for 300 s. After each cycle, the sensor chip was regenerated by injecting 10 mM glycine (pH 1.5) for a duration of 90 s. The binding data were determined by subtracting the background signal from both the blank cycle and the reference flow cell. The binding and dissociation curves of nanobodies at different concentrations to RBD were plotted by fitting the data according to the 1:1 binding model.

### 2.8. Competitive Phage ELISA for Epitope Grouping

WT RBD protein (50 ng/well) was coated onto a 96-well microplate overnight at 4 °C. Subsequently, 300 μL/well of blocking solution was added to the microplate for blocking. After three PBST rinses, the 1:1 mixture samples of phage-displayed nanobody supernatants (diluted 1:10) and purified nanobodies (10 μg/mL) were added to the wells, with an addition volume of 100 μL per well. Additionally, control wells were set up without purified nanobodies for noncompetitive binding. Following incubation at 37 °C for one hour, subsequent experiments were carried out as described above using HRP-conjugated anti-M13 antibody, TMB substrate, stop solution and a plate reader.

### 2.9. RBD-ACE2 Binding Inhibition Assayed by Competitive ELISA

Briefly, WT RBD protein (50 ng/well) was coated onto a 96-well microplate overnight at 4 °C. After blocking, the 1:1 mixture samples of purified nanobodies (1 μg/mL or 10 μg/mL) and biotinylated ACE2 (0.4 μg/mL) were added to the wells, with an addition volume of 100 μL per well. Additionally, control wells were set up without purified nanobodies for noncompetitive binding. Following incubation at 37 °C for one hour, subsequent experiments were carried out as described above using HRP-conjugated streptavidin antibody (1:10,000, Proteintech), TMB substrate, stop solution and a plate reader.

### 2.10. Pseudovirus Neutralization Assay

To generate SARS-CoV-2 pseudovirus, HEK293T cells were seeded in 6-well plates the day prior to transfection. On the second day, cells were cotransfected with three plasmids, including one encoding the spike glycoprotein of SARS-CoV-2 WT, BA.1, BA.4/5, BQ.1.1, XBB.1 or single mutants along with psPAX2 and pLenti-GFP. After a six-hour transfection period, the culture medium was replaced with fresh DMEM containing 10% FBS. After a 48 h transfection period, the supernatants containing pseudovirus were collected, and any cellular residue was removed through filtration using 0.45 µm syringe filters (Merck, Billerica, MA, USA).

For the pseudovirus neutralization assay, nanobodies were serially diluted threefold and added to 96-well white opaque plates (Beyotime) at a volume of 50 μL per well. Afterward, 50 μL per well of pseudovirus solution was added to the plates and mixed with nanobodies, followed by incubation at 37 °C for one hour. The wells for the virus control contained only the pseudovirus solution. Next, 2 × 10^4^ HEK293-hACE2 cells were seeded onto white opaque plates and incubated at 37 °C with 5% CO_2_ for a duration of 48 h. The white opaque plates were equilibrated at ambient temperature for 10 min before adding 100 μL of firefly luciferase assay reagent (Beyotime) to each well. After a 5 min incubation, chemiluminescence detection on a SpectraMax i3x plate reader was utilized to quantify the intracellular fluorescence signal of each well. The percent neutralization was calculated by comparing it with the virus control. Then, 4-parameter logistic regression was utilized to determine the IC_50_ values. The experiments were performed in triplicate and repeated two or more times.

### 2.11. Stability Analysis

The in vitro stability of multivalent nanobodies was assessed by subjecting them to storage conditions at 4 °C, room temperature (RT), or 37 °C for either two or four weeks. Subsequently, their binding affinity towards the Omicron XBB.1 S-trimer protein was measured using ELISA.

To determine the thermal stability, multivalent nanobodies were incubated in a water bath at temperatures of 25, 37, 50, 60, 70, 80 and 90 °C for half an hour and then equilibrated to room temperature. All treated multivalent nanobodies were diluted to a concentration of 8 ng/mL, and their binding affinity toward the Omicron XBB.1 S-trimer protein was measured using ELISA. The relative binding activity was determined by calculating the ratio of OD_450_ values after heat treatment to OD_450_ values before treatment and then multiplying the results by 100%.

### 2.12. Statistical Analysis

All statistical analyses were performed with GraphPad Prism software (version 8.0). Data are presented as the mean ± SD of at least two replicates. The 4-parameter logistic regression analysis was utilized to determine the EC_50_ and IC_50_ values.

## 3. Results

### 3.1. Construction and Screening of a Nanobody Phage Display Library

For the rapid identification of RBD-specific nanobodies with high affinity and appropriate diversity, we sought to construct a highly diverse nanobody phage display library (Figure 1). Initially, we administered six rounds of immunization to four alpacas using SARS-CoV-2 S1 and RBD proteins. A total of 5 μg of RNA with satisfactory integrity was extracted from 2 × 10^7^ PBMCs obtained from immunized alpacas, which served as the template for cDNA synthesis (Appendix A). Then, the VHH genes were amplified using nested PCR. In the first round of PCR, the 1000 bp (VH-CH1-CH2) and 750 bp (VHH-CH2) gene fragments were amplified using cDNA as a template (Appendix A). The 750 bp gene fragments were subsequently used as templates for amplification in the next round of PCR, resulting in the generation of VHH gene fragments approximately 400 bp in length (Appendix A). The VHH genes were then cloned into the pHEN1 vector, followed by the transformation of TG1 cells, resulting in a VHH library of ~1.3 × 10^7^ cfu (Appendix A). The validation of 24 randomly picked clones was conducted through colony PCR, revealing a 100% insertion rate for the VHH library (Appendix A). All 24 clones were subsequently sequenced. A total of 23 clones contained the correct framework regions of VHH genes and had different complementary determining region (CDR) compositions, indicating that our established library had a 96% in-frame rate and 100% VHH gene diversity (Appendix A). Overall, the above results suggest that the VHH library was constructed with high quality and satisfactory diversity. Then, the rescue of this VHH library with M13KO7 helper phages yielded a nanobody phage display library with a size of 4.4 × 10^14^ pfu/mL (Appendix A).

To isolate potential RBD-specific binders from the nanobody phage display library, we used the RBD protein as bait in three rounds of biopanning (Figure 1). The enrichment factor of each round of biopanning was calculated, and the results demonstrate the effective enrichment of RBD-binding phages (Figure 2A). Afterward, 106 clones were randomly picked from the third round of biopanning and assessed for their binding to the WT RBD using phage ELISA. Remarkably, 101 out of 106 clones were identified as positive with binding ratios higher than 3 (Figure 2B), and these were sent for sequencing. Sequence analysis revealed that a total of 26 unique nanobodies were obtained based on the classification of amino acid sequences in the CDRs (Figure 2C). Phylogenetic tree analysis showed the sequence diversity of these nanobodies (Figure 2D). Based on the sequencing and phage ELISA results, nine nanobodies (named A14, A31, B13, B18, B20, B27, C2, C8, and D11) were chosen as candidates for further expression and purification. SDS–PAGE analysis showed that these nine nanobodies were isolated with high purity, and their molecular weight was approximately 15 kDa, which is consistent with the theoretical value (Figure 2E).

### 3.2. Characterization and Epitope Grouping of the Nanobody Candidates

We first investigated the binding ability of the nine nanobody candidates to both the WT RBD and BA.1 RBD using ELISA. The results demonstrate that all nanobodies bound strongly to the WT RBD with EC_50_ values ranging from 0.26 to 23.8 ng/mL (Figure 3A). Compared with ACE2, most of the nanobodies (except for B27) displayed stronger positive signals with EC_50_ values lower than 0.9 ng/mL (Figure 3A). Strikingly, we found that most nanobodies (except for B27) still exhibited strong binding to BA.1 RBD with EC_50_ values below 107 ng/mL (Figure 3B). Nevertheless, except for C8, all nanobodies showed lower binding activities to BA.1 RBD compared to WT RBD. Further surface plasmon resonance (SPR) experiments revealed that most nanobodies (except for B27) bound to the WT RBD with high affinity (Table 1 and Appendix A). The binding affinities (KD) ranged from 0.013 to 0.775 nM (Table 1). Consistent with the ELISA results, most nanobodies (seven out of eight) exhibited relatively weak binding affinity with Omicron BA.1 RBD, but their KD values (except for B27) remained in the single-digit nanomolar or lower range (Table 1 and Appendix A). Notably, A14 and B20 showed strong binding affinity to Omicron BA.1 RBD with KD values of 0.065 nM and 0.026 nM, respectively (Table 1). Unfortunately, the binding affinity of C8 was not measured successfully by SPR, presumably owing to the inapplicability of the CM5 chip.

To explore whether these nine nanobodies bind to distinct epitopes, their competitive ability for RBD binding was assessed using a competitive phage ELISA. Purified nanobodies at high concentrations (up to 5 μg/mL) were used to compete with phage-displayed nanobodies to bind the WT RBD. The competition is summarized in Figure 3C. A residual binding of a competing pair of less than 20% suggests that these two nanobodies may target the same or similar epitopes. Conversely, a residual binding of a competing pair that was greater than 90% suggests that these two nanobodies recognize nonoverlapping epitopes on the RBD. We found that these nine nanobodies were highly diverse, as they could be classified into five competition groups, indicating the recognition of five nonoverlapping epitopes on the RBD (Figure 3C). Subsequently, the blocking activities of these nanobodies against RBD-ACE2 were assessed through competition ELISA. As shown in Figure 3D, ACE2 binding to the RBD was effectively blocked by A14 and D11 at an excess concentration of 5 μg/mL, with a blocking rate greater than 50%. The other nanobodies exhibited low blocking activities at 5 μg/mL and did not demonstrate any blocking effect at 0.5 μg/mL. These results suggest that the epitopes of A14 and D11 may partly overlap with the RBM (also known as the ACE2-binding site), while the epitopes bound by the remaining nanobodies (A31, B13, B18, B20, B27, C2 and C8) are likely distant from the RBM. Overall, these experiments indicate that these nine nanobodies targeting five nonoverlapping epitopes on the RBD showed satisfactory cross-binding activity against the WT RBD and BA.1 RBD, thereby establishing a robust foundation for the development of cross-reactive neutralizing antibodies.

### 3.3. Cross-Neutralization Capacity of Nanobodies against Diverse SARS-CoV-2 Omicron Subvariants

During the characterization of these nanobodies, Omicron subvariants BA.1, BA.4/5, BQ.1.1 and XBB.1 successively became dominant in numerous regions globally, and were found to evade most available neutralizing antibody responses [6,8,13]. To evaluate the cross-neutralization capacity of the nanobodies toward these Omicron subvariants, we conducted lentivirus-based pseudovirus infection assays. Initially, we successfully packaged five pseudoviruses harboring the spike glycoprotein of SARS-CoV-2 WT, BA.1, BA.4/5, BQ.1.1, or XBB.1. The neutralizing efficacy of the nine nanobodies against the indicated SARS-CoV-2 pseudoviruses was quantified at a fixed concentration of 5 μg/mL (Figure 4A). A neutralization efficiency below 50% was defined as nonneutralizing. We observed that neither B20 nor C8 could neutralize any of the tested pseudoviruses, even at a high concentration of 5 μg/mL. However, they exhibited strong binding affinities towards both WT RBD and BA.1 RBD. These results suggest that B20 and C8 specifically target nonneutralizing epitopes on the RBD.

Then, the nanobodies with efficiencies higher than 50% were serially diluted threefold. The neutralization curves against each pseudovirus are presented in Figure 4B–F, and the IC_50_ values are shown in Figure 4G. Encouragingly, A14 in group C exhibited cross-protective activities against all pseudoviruses, with IC_50_ values ranging from 83.93 to 316.6 ng/mL, thereby indicating its potential for targeting a conserved RBD epitope. (Figure 4G). Three nanobodies (A31, B13, and B18) in group A demonstrated effective neutralizing activity against SARS-CoV-2 WT, BA.1, and BA.4/5 pseudoviruses with IC_50_ values ranging from 117.6 to 1026 ng/mL; however, C2 in group B exhibited a limited neutralizing capacity against them (IC_50_ > 2 µg/mL) (Figure 4G). Additionally, D11 effectively neutralized both SARS-CoV-2 WT and Omicron BA.1 pseudoviruses, whereas B27 exclusively neutralized the WT pseudovirus.

In total, we obtained five nanobodies (A14, A31, B13, B18, and C2) that specifically target three nonoverlapping epitopes (A, B, and C) on the RBD and effectively neutralize Omicron BA.1 and BA.4/5. Among them, A14 maintained neutralizing activity against Omicron BQ.1.1 and XBB.1. However, they all lacked the necessary potency for therapeutic application, and required further optimization through a multimerization strategy to enhance their potency and broad-spectrum neutralization. Due to being in the same competition group, B18 exhibited a relatively weak neutralization potency compared to A31 and B13; therefore, it was not considered for further research. Finally, A14, A31, B13, and C2 were chosen as modular units for further multivalent engineering.

### 3.4. Generation of Diverse Multivalent Nanobodies with Enhanced Neutralizing Potency and Breadth

To assess the potential of multivalency in enhancing the neutralizing potency and breadth of A14, A31, B13, and C2 monomers, we designed nine multivalent nanobodies comprising homo- or heterodimeric nanobodies as well as homo- or heterotrimeric nanobodies (Figure 5A). These multivalent nanobodies were prepared by connecting two or three nanobody monomers through optimized flexible GlySer linkers of 20 amino acids [(G_4_S)_4_]. We found that they could be easily expressed in BL21 cells with purification yields of approximately 20–50 µg per milliliter in shake flask culture. SDS–PAGE analysis showed that the dimeric (≈30 kDa) and trimeric (≈45 kDa) nanobodies exhibited high levels of purity and had the expected molecular weights (Figure 5B).

Next, we assessed the neutralizing efficacy of the generated multivalent nanobodies against SARS-CoV-2 WT, BA.1, BA.4/5, BQ.1.1, and XBB.1 pseudoviruses (Figure 5C) and presented the corresponding IC_50_ values (Figure 5D). Surprisingly, these multivalent nanobodies, except for A31-A31, were able to neutralize all tested pseudoviruses. The B13-B13-B13 homotrimer exhibited the most potent neutralizing activities, with a GM IC_50_ value of 20.83 ng/mL (Figure 5D). Its neutralizing potency was significantly enhanced compared to that of the B13 monomer, with improvements ranging from >86.1- to 268-fold against all tested pseudoviruses. The homodimers A31-A31 and B13-B13, as well as the heterodimers A31-C2 and B13-C2, exhibited potent neutralization activity against SARS-CoV-2 WT, BA.1 and BA.4/5 with IC_50_ values below 24 ng/mL (Figure 5D). Similarly, their neutralizing potency increased significantly by approximately 15.4–113-fold compared to the respective monomers A31 and B13. Notably, the heterodimers A31-C2 and B13-C2 exhibited superior efficacy against Omicron BQ.1.1 and XBB.1 in comparison to the homodimers A31-A31 and B13-B13. Although four multivalent nanobodies encompassing the A14 modular unit (A14-A31, A14-B13, A14-C2, and A14-A31-C2) exhibited broad neutralization against all five tested pseudoviruses, their improvement in neutralization efficacy compared to monomer A14 was minimal or not significant (Figure 5D). Taken together, these data indicate that the multimerization strategy can enhance the neutralizing potency and breadth of monomeric nanobodies.

### 3.5. Analysis of the Mechanism Underlying the Enhancement of Neutralization Breadth by Multivalent Nanobodies

To further explore the mechanism underlying the enhanced neutralizing breadth of multivalent nanobodies, we selected B13-B13, B13-C2, and B13-B13-B13 as representative antibodies for subsequent investigation. Initially, we examined whether these three multivalent nanobodies exhibited higher binding affinities toward Omicron BQ.1.1 and XBB.1 S-trimer than their monomeric counterparts. As expected, the binding affinities of B13-C2, B13-B13, and B13-B13-B13 to the Omicron BQ.1.1 S-trimer protein showed 643-fold, 482-fold and 965-fold increases, respectively, compared to that of B13 (Figure 6A). Similarly, their binding to the Omicron XBB.1 S-trimer protein was 157-fold, 94-fold, and 471-fold tighter than that of B13 (Figure 6A). Compared to C2, the binding activities of B13-C2 to the Omicron BQ.1.1 and XBB.1 S-trimer proteins were also increased by 12.2-fold and 15.7-fold, respectively (Figure 6A). These results suggest that the joint structure of our multivalent nanobodies can significantly enhance the binding affinity of monomeric nanobodies to the Omicron S-trimer, thereby augmenting their neutralization capacity.

Compared to BA.4/5, the RBD proteins of BQ.1.1 and XBB.1 contain an additional seven mutations [14]. To validate the influence of these additional mutations, we constructed seven mutated pseudoviruses based on BA.4/5 to identify the crucial residues responsible for conferring resistance to B13 and C2. The neutralization curves and IC_50_ values of B13 and C2 against seven mutated pseudoviruses are presented in Figure 6B,C. Compared with BA.4/5, B13 exhibited a significant reduction in its neutralizing capacity against the R346T mutated pseudovirus, with an IC_50_ reduction of more than 10.8-fold. Moreover, the neutralizing activity of monomer C2 against the N460K mutated pseudovirus decreased, with an IC_50_ > 5 µg/mL (the fold change cannot be calculated). The co-occurrence of the R346T and N460K mutations in BQ.1.1 and XBB.1 accounts for the resistance of these two subvariants to neutralization by B13 and C2 (Figure 4G). However, in sharp contrast, B13-C2, B13-B13, and B13-B13-B13 multivalent nanobodies demonstrated remarkable neutralization breadth and enhanced potency (Figure 6B,C). They potently neutralized all seven single mutants, with IC_50_ values below 100 ng/mL. Thus, the R346T and N460K mutations enable the virus to evade monomeric but not multivalent nanobodies. Notably, there was a slight decrease (3.1–4.4-fold) in the neutralization activities of these three multivalent nanobodies toward the R346T- and/or N460K-mutated pseudoviruses (Figure 6C). These findings elucidate the reasons behind the reduced neutralization activity of these multivalent nanobodies against BQ.1.1 and XBB.1 in comparison to BA.4/5 (Figure 5D). In short, our multivalent nanobodies (B13-C2, B13-B13, and B13-B13-B13) with enhanced neutralization potency could accommodate the R346T and N460K escape mutations present on the Omicron BQ.1.1 and XBB.1 subvariants, resulting in an extraordinary breadth of neutralization.

### 3.6. Excellent Stability of Multivalent Nanobodies

Among dimeric and trimeric nanobodies, B13-C2 and B13-B13-B13 exhibited the highest neutralizing activity, respectively. Therefore, we selected them as representative antibodies for further stability analyses. The in vitro stabilities of B13-C2 and B13-B13-B13 were assessed by storing them at different temperatures for either two or four weeks, followed by measuring their binding affinity towards the Omicron XBB.1 S-trimer protein using ELISA. As shown in Figure 7A, the binding capacity of both B13-C2 and B13-B13-B13 remained almost unchanged when stored at three different temperatures for up to 2 weeks. While being stored at room temperature or 37 °C for 4 weeks, their binding affinity showed slight degradation but remained within the single-digit ng/mL range. Furthermore, B13-C2 and B13-B13-B13 exhibited excellent thermal stability (Figure 7B,C). After incubation at 70 °C for 30 min, the relative binding activities of both B13-C2 and B13-B13-B13 remained above 90%. However, their relative binding activities significantly degraded after being incubated at 80 °C for 30 min, while still remaining above 50%. The above results demonstrate that our engineered multivalent nanobodies exhibit remarkable drug stability.

## 4. Discussion

The ongoing antigenic evolution of SARS-CoV-2 significantly hampers the efficacy of neutralizing antibody therapies. Omicron subvariants exhibit resistance against most existing neutralizing antibodies due to extensive spike mutations [4,5,6,7,14,46]. Therefore, it is imperative to develop potent and broadly neutralizing antibodies capable of effectively targeting the Omicron subvariants as well as future emerging variants.

Numerous studies have confirmed the difficulty of monotherapy in persistently resisting the incessantly emerging mutants of SARS-CoV-2. Multivalent engineering has been employed as an effective strategy for enhancing the neutralizing potency of antibodies in combating SARS-CoV-2 variants [47,48,49,50]. Moreover, in terms of cost-effectiveness and efficacy, multivalent antibodies offer a distinct advantage over antibody cocktails [48,49]. Nanobodies, which consist of a single Ig domain, offer easier engineering into homo/heteromultimers compared to conventional antibodies [51]. Additionally, their small size allows them to bind conserved epitopes that are not typically targeted by monoclonal antibodies [52]. These characteristics make nanobodies highly appealing during the COVID-19 pandemic. Although numerous monomeric nanobodies and multivalent nanobodies have been reported to effectively neutralize SARS-CoV-2, limited data are available on their neutralizing activities against multiple Omicron subvariants, particularly BQ.1.1 and XBB.1 [42,43,53,54,55,56].

Several prior studies have demonstrated the presence of conserved epitopes on the RBD, whether in the RBM region or the non-RBM region, highlighting the potential to use the RBD as an effective target for developing potent and broadly neutralizing antibodies [33,34,35,36,57,58]. In this study, our aim was to initially develop robust nanobodies that target diverse epitopes on the RBD and effectively inhibit various Omicron subvariants. A highly diverse nanobody phage display library can increase the possible diversity of epitopes [59]. Therefore, when generating a phage library, we amplified the VHH genes using multiple primers to introduce CDR diversity. After three rounds of biopanning, nine nanobodies that tightly bind to WT RBD and BA.1 RBD were identified. Based on epitope grouping analysis, these nanobodies can be classified into five competition groups (A–E). A pseudovirus neutralization assay demonstrated that only nanobodies from groups A–C effectively neutralized the SARS-CoV-2 WT strain. Four nanobodies (A31, B13, B18, and C2) from groups A and B showed effective neutralizing activity against subvariants BA.1 and BA.4/5, but completely lost their ability to neutralize subvariants BQ.1.1 and XBB.1, suggesting that they target nonconserved epitopes on the RBD. Encouragingly, A14 from group C demonstrated broad-spectrum neutralization against all tested Omicron subvariants, suggesting that it may target a conserved epitope on the RBD. Notably, the discovery of the broad-spectrum neutralizer A14 has demonstrated that immunization with prototype SARS-CoV-2 antigens can also induce broadly neutralizing antibodies targeting conserved epitopes [60].

Currently, most of the available RBD-targeting neutralizing antibodies directly or sterically inhibit ACE2 binding to the S protein, thereby blocking viral entry. Similarly, A14 showed moderate ACE2-RBD blocking activity in the competition ELISA, indicating that its primary mechanism of neutralization is ACE2 blockade. However, all nanobodies from groups A and B, except D11, were unable to inhibit ACE2-RBD binding. This suggests that their binding epitopes are located outside the RBM. The neutralization mechanism of these non-RBM-targeting nanobodies needs to be further studied, possibly by destabilizing the SARS-CoV-2 spike trimer or blocking potential syncytia formation [30,31]. In general, we obtained three groups of RBD-targeting neutralizing nanobodies with diverse epitopes and different neutralization mechanisms, which has established a robust foundation for the development of potent and broadly multivalent nanobodies.

Given the natural trimeric presentation of spike proteins on the surface of SARS-CoV-2, the multimerized nanobody has the potential to simultaneously target multiple RBD antigens, thereby yielding profound gains in potency [54]. The multimeric designs primarily include homodimers, heterodimers, homotrimers, and heterotrimers. Therefore, focusing on the aforementioned four designs, we selected four neutralizing nanobodies (A31 and B13 from group A, C2 from group B, and A14 from group C) to construct twelve diverse multivalent nanobodies. However, due to the low yield of protein expression for three multivalent nanobodies (A14-A14, A14-A14-A14, and A31-A31-A31), we were only able to obtain nine purified multivalent nanobodies. As expected, most multivalent nanobodies displayed significant improvement against Omicron subvariants compared to monomers. Notably, two homodimers (A31-A31, B13-B13), two heterodimers (A31-C2, B13-C2), and one homotrimer (B13-B13-B13) showed high neutralizing potency toward SARS-CoV-2 WT and Omicron BA.1 and BA.4/5, with IC_50_ values below 24 ng/mL. In addition to A31-A31, they also effectively neutralized Omicron BQ.1.1 and XBB.1, although the corresponding monomers (A31, B13, and C2) individually failed to do so. Moreover, we found that the heterodimers A31-C2 and B13-C2 were more effective against BQ.1.1 and XBB.1 than the homodimers A31-A31 and B13-B13. The results indicate that the fusion of nanobodies that target distinct epitopes into heterodimeric forms represents a promising strategy to overcome escape mutants, aligning with previous research findings [61,62]. We also observed that the in vitro neutralization potencies of nanobodies against pseudoviruses can gradually increase with valency. This is exemplified by B13, which showed 109-, 120-, 268-, >146-, and >86.1-fold increases in the neutralization of SARS-CoV-2 WT, Omicron BA.1, BA.4/5, BQ.1.1, and XBB.1 pseudoviruses, respectively, from B13 to B13-B13 to B13-B13-B13. The other four multivalent nanobodies (A14-A31, A14-B13, A14-C2, and A14-A31-C2) showed a smaller increase in neutralization potency compared to the corresponding A14 monomer, which may have resulted from the spatial constraints of the linker [63]. Significantly, in this study, the most potent multivalent nanobody, B13-B13-B13, outperformed previously reported broadly neutralizing antibodies, such as S728-1157 and VacBB-551 [33,64]. Our studies have demonstrated that the rational engineering of multivalent nanobodies offers a rapid approach to generate potent viral neutralizers with enhanced efficacy in controlling viral escape mutants.

Determining the binding affinity between our multivalent nanobodies and the S-trimer proteins of Omicron BQ.1.1 and XBB.1 is undoubtedly crucial for elucidating their enhanced neutralization breadth. The ELISA demonstrated that the S-trimer binding affinities of B13-C2, B13-B13, and B13-B13-B13 multivalent nanobodies to BQ.1.1 and XBB.1 were significantly higher than those of B13 and C2, which is basically consistent with the neutralization results. The RBD proteins of BQ.1.1 and XBB.1 subvariants exhibit seven additional mutations (R346T, L368I, K444T, V445P, G446S, N460K, and F490S) compared to the BA.4/5 subvariant [14]. These extra mutations contribute to the enhanced immune evasion of BQ.1.1 and XBB.1 against therapeutic antibodies [65]. The BQ.1.1 subvariant exhibited increased resistance to neutralization, mainly due to the crucial N460K mutation, with the R346T and K444T mutations also having some impact [66]. Another study demonstrated that the R346X mutation in the RBD, mainly R346T, provides additional fitness to Omicron subvariants, such as XBB and BQ, which might facilitate antibody evasion [67]. In our study, the neutralizing capacity of B13 was significantly abolished by the R346T mutation, while the reduction in neutralizing activity of C2 was attributed to the N460K mutation. However, B13-C2, B13-B13, and B13-B13-B13 multivalent nanobodies exhibited remarkable efficacy in neutralizing R346T and N460K mutants owing to their rational design for multimerization. Notably, the R346T mutation did not confer resistance to B13-B13-B13. The neutralizing efficacy of the B13-C2 heterodimer against BQ.1.1 and XBB.1 was impacted by a single mutation (R346T), while the B13-B13 homodimer was affected by two mutations (R346T and N460K). This explains why B13-B13 have weaker efficacy than B13-C2. These findings reconfirm that the conjoining of two distinct paratopes is preferred to ensure improved resistance against viral escape. Collectively, the multimerization of nanobodies can enlarge the interface area, enhance the affinity between an individual nanobody and spike protein, and enable better toleration of escape mutations, which may account for the mechanism of the broad neutralization mediated by our multivalent nanobodies.

Recently, Yao et al. reported a homotrimer, Nb4-16t, that neutralizes multiple Omicron variants, including BQ.1 and XBB.1 [68]. However, Nb4-16t was engineered using a monomeric nanobody (Nb4) that specifically targets a conserved epitope, which differs from our design strategy. In our study, we selected the monomer B13, which targets a nonconserved epitope, to construct the homotrimer B13-B13-B13, which can also effectively neutralize Omicron BA.1, BA.4/5, BQ.1.1, and XBB.1 at low concentrations. More importantly, we also observed a similar effect when utilizing monomers B13 and C2 to engineer heterodimer B13-C2, despite the weak potency of C2 (IC_50_ > 2 µg/mL). To our knowledge, a heterodimer similar to B13-C2 with potent neutralizing activities against Omicron BQ.1.1 and XBB.1 has not been reported thus far. Most researchers prefer constructing multimers of potent neutralizing nanobodies that target conserved epitopes, which has also proven to be an effective strategy [55,68,69]. Our study demonstrated that even if nanobodies lose their neutralization activity against new SARS-CoV-2 variants due to the targeting of nonconserved epitopes, they may still be selected for developing multivalent nanobodies with enhanced potency and neutralization breadth. This finding could enhance researchers’ understanding of nanobody multimerization strategies, thereby transforming previously developed partial nanobodies into a formidable arsenal to help combat the pandemic.

The manufacturability and efficient large-scale production of antibodies is crucial in addressing pandemics such as COVID-19. Under laboratory conditions, our engineered multivalent nanobodies can be easily produced with a high yield and demonstrate exceptional stability. These characteristics suggest their potential for rapid and widespread utilization as therapeutic agents against COVID-19. Moreover, in comparison to the antibody combinations used in cocktail therapy, our engineered multivalent nanobodies are single molecules that integrate two or three nanobodies, thereby exhibiting high manufacturability. In contrast with conventional monoclonal antibodies, multivalent nanobodies offer distinct advantages. Most importantly, their small size enables direct delivery to the site of infection through inhalation [70]. Additionally, because of the lack of the Fc domain, multivalent nanobodies could prevent the emergence of antibody-dependent enhancement (ADE) and thus represent a safer alternative to monoclonal antibodies [71]. In our upcoming study, further investigations are needed to uncover the structural basis of multivalent nanobodies neutralizing multiple Omicron subvariants. In addition, the efficacies of these multivalent nanobodies against the other Omicron subvariants need further evaluation. Finally, the protective efficacy of inhalation of the multivalent nanobody B13-B13-B13 deserves further investigation.

## 5. Conclusions

In summary, we identified nine RBD-specific nanobodies from a high-diversity VHH phage display library. Based on the epitope grouping and neutralization assay data, we designed and engineered a set of multivalent nanobodies that exhibit enhanced potency and breadth in neutralization. The development of these multivalent nanobodies expands the arsenal available for combating SARS-CoV-2 variants. Furthermore, the nanobody multimerization strategy is expected to be useful in developing cost-effective and more efficacious antibody therapies for future pandemics caused by similar viruses.

## Figures and Tables

**Figure 1 vaccines-12-00417-f001:**
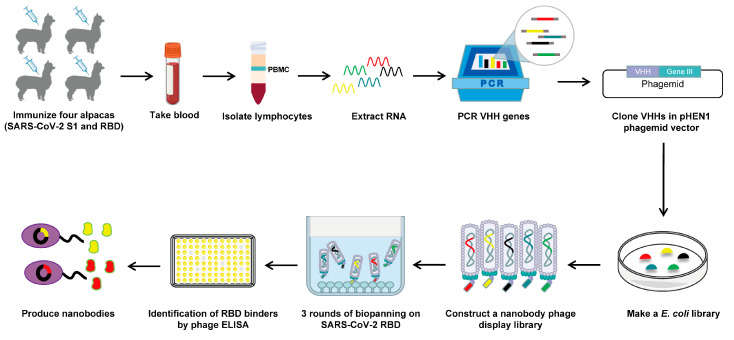
Schematic depicting the construction of a nanobody phage display library and the screening strategy used.

**Figure 2 vaccines-12-00417-f002:**
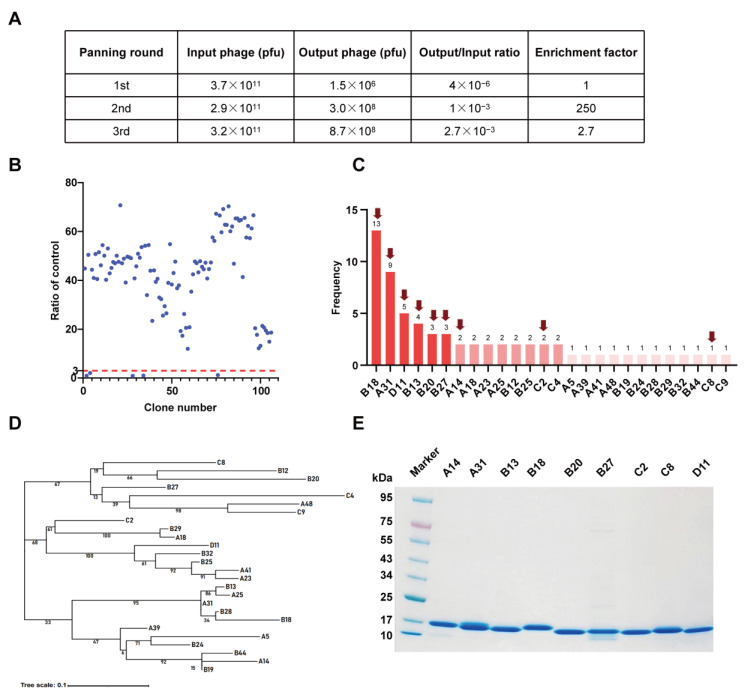
Identification of SARS-CoV-2 RBD-specific nanobodies. (**A**) Enrichment of phage particles binding to the RBD through three rounds of biopanning. (**B**) Phage ELISA for the identification of positive clones. A binding ratio higher than 3 compared to the negative control was considered positive. (**C**) According to the sequencing results, 26 unique nanobody sequences were identified, and the frequency of each sequence is displayed. The nanobodies indicated by the red arrow were chosen as candidates. (**D**) Phylogenetic tree showing the sequence diversity of 26 unique nanobodies. (**E**) The SDS–PAGE gel showing the sizes and purity of the nanobodies.

**Figure 3 vaccines-12-00417-f003:**
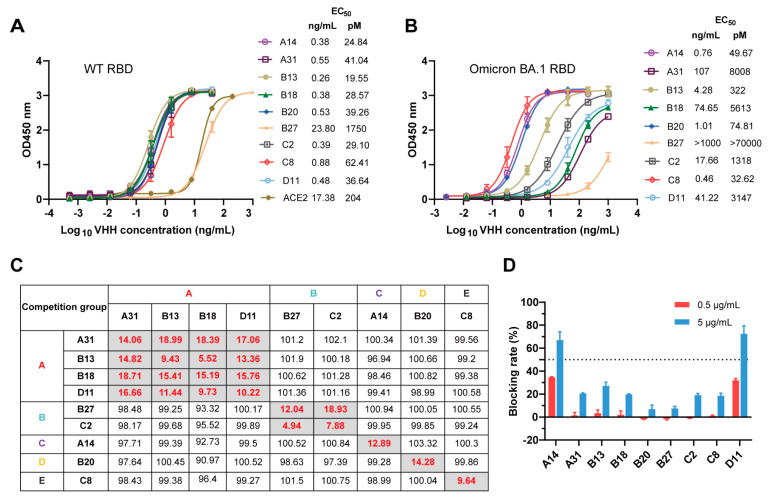
Characterization and epitope grouping of the nanobody candidates. (**A**,**B**) Binding activities of the nanobodies and ACE2 to WT RBD (**A**) and BA.1 RBD (**B**), evaluated by ELISA. (**C**) Competition of these nanobodies with each other for WT RBD binding was measured by competitive phage ELISA. The left column and top row show the purified nanobodies and phage-displayed nanobodies, respectively. The nanobodies are displayed in five competition groups (A, B, C, D, or E). The values in the boxes represent the residual binding of a competing pair, which was calculated as the percentage of competitive binding relative to noncompetitive binding and standardized to 100%. The values in gray boxes are below 20%, and the values in white boxes exceed 90%. (**D**) RBD-ACE2 blocking activities of nanobodies characterized by competition ELISA.

**Figure 4 vaccines-12-00417-f004:**
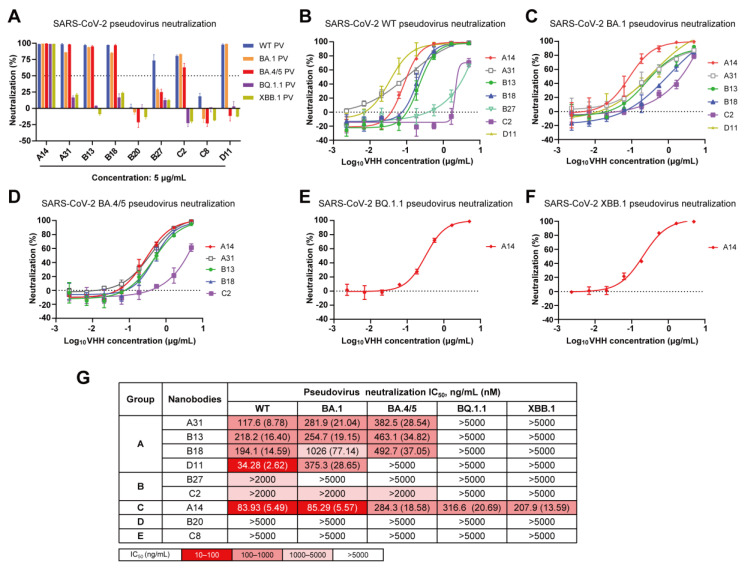
Cross-neutralization capacity of nanobodies against SARS-CoV-2 WT, BA.1, BA.4/5, BQ.1.1 and XBB.1 pseudoviruses. (**A**) Neutralization efficiency of nine nanobodies against five pseudoviruses at a concentration of 5 µg/mL. (**B**–**F**) Neutralization curves of nanobodies against SARS-CoV-2 WT (**B**), Omicron BA.1 (**C**), BA.4/5 (**D**), BQ.1.1 (**E**) and XBB.1 (**F**) pseudoviruses. (**G**) Summary of IC_50_ values of nanobody pseudovirus neutralization.

**Figure 5 vaccines-12-00417-f005:**
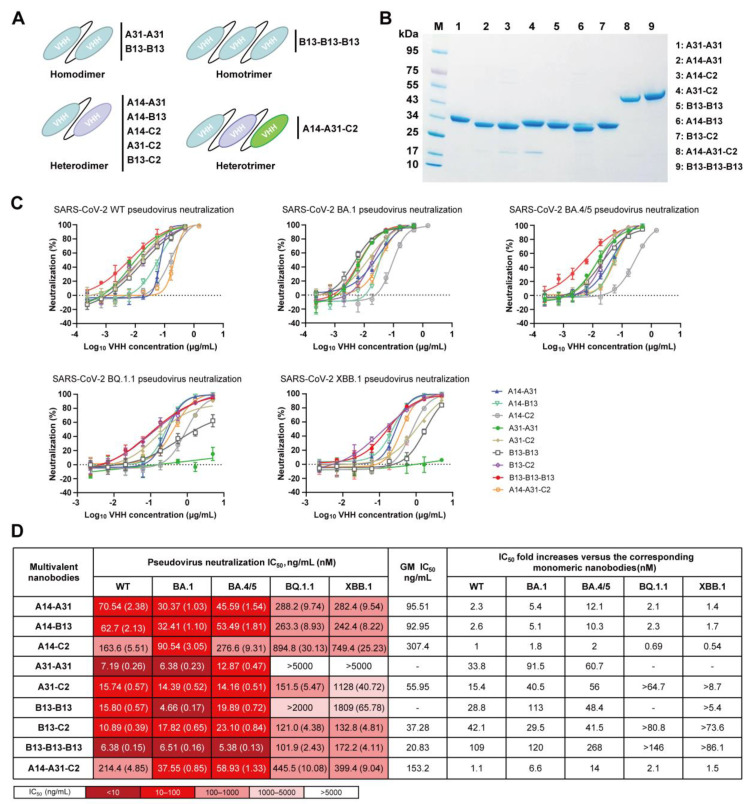
Neutralizing activity of multivalent nanobodies against SARS-CoV-2 WT, BA.1, BA.4/5, BQ.1.1 and XBB.1 pseudoviruses. (**A**) Schematic diagram of engineered multivalent nanobodies, including homo- or heterodimers and homo- or heterotrimers. (**B**) The SDS–PAGE gel showing the sizes and purity of the multivalent nanobodies. (**C**) Neutralization curves of multivalent nanobodies against five pseudoviruses. (**D**) Summary of IC_50_ values of multivalent nanobody pseudovirus neutralization. IC_50_-fold increases in multivalent nanobodies versus the corresponding monomeric nanobodies were calculated. “-” indicates that fold changes cannot be calculated.

**Figure 6 vaccines-12-00417-f006:**
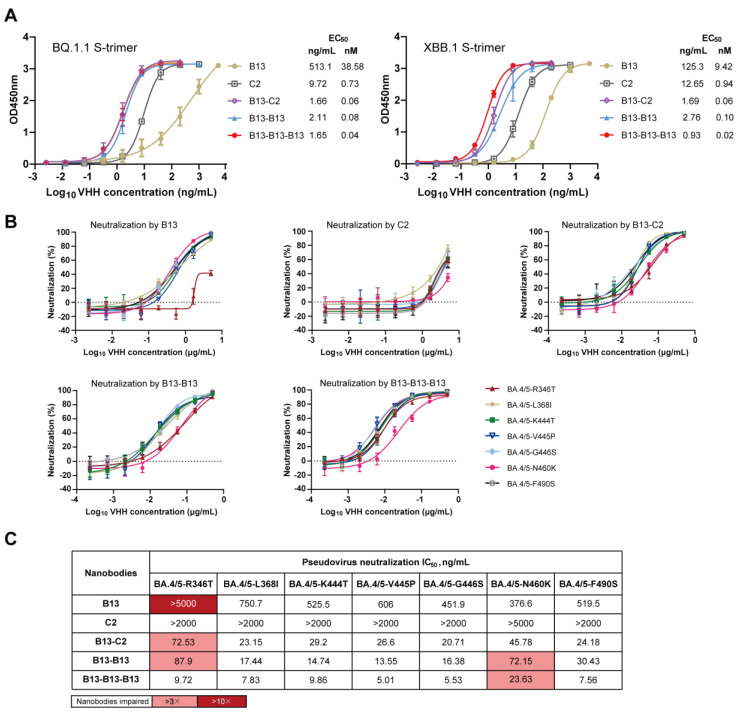
The mechanism analysis of B13-C2, B13-B13, and B13-B13-B13 with enhanced neutralizing breadth. (**A**) Binding activity of B13, C2, B13-C2, B13-B13, and B13-B13-B13 to S-trimer proteins of Omicron BQ.1.1 and XBB.1. (**B**) Neutralization curves of B13, C2, B13-C2, B13-B13, and B13-B13-B13 against seven single mutants of the Omicron BA.4/5. (**C**) The IC_50_ values of B13, C2, B13-C2, B13-B13, and B13-B13-B13 against seven single mutants. Their respective fold decreases in IC_50_ relative to BA.4/5 were calculated. Fold changes > 3 are highlighted in pink, and those >10 are highlighted in red.

**Figure 7 vaccines-12-00417-f007:**
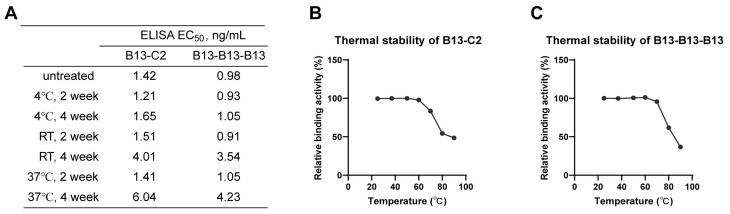
Stability analysis of B13-C2 and B13-B13-B13 multivalent nanobodies. (**A**) The in vitro stability of B13-C2 and B13-B13-B13. RT indicates room temperature. (**B**,**C**) Thermal stability of B13-C2 and B13-B13-B13.

**Table 1 vaccines-12-00417-t001:** The binding affinities of nanobodies to different RBDs.

Nanobodies	WT RBD			Omicron BA.1 RBD		
	k_on_ (1/Ms)	k_off_ (1/s)	KD (nM)	k_on_ (1/Ms)	k_off_ (1/s)	KD (nM)
A14	1.016 × 10^6^	1.364 × 10^−5^	0.013	6.732 × 10^5^	4.350 × 10^−5^	0.065
A31	1.392 × 10^6^	3.574 × 10^−4^	0.257	1.421 × 10^5^	7.790 × 10^−4^	5.484
B13	4.198 × 10^4^	6.378 × 10^−7^	0.015	1.090 × 10^5^	2.920 × 10^−5^	0.268
B18	1.008 × 10^6^	2.300 × 10^−4^	0.228	2.177 × 10^5^	6.819 × 10^−4^	3.132
B20	1.580 × 10^6^	1.196 × 10^−4^	0.076	1.095 × 10^7^	2.884 × 10^−4^	0.026
B27	2.667 × 10^5^	6.329 × 10^−3^	23.73	9.800 × 10^4^	5.868 × 10^−3^	59.88
C2	1.534 × 10^5^	1.189 × 10^−4^	0.775	1.284 × 10^5^	1.428 × 10^−4^	1.113
C8	N.D.	N.D.	N.D.	N.D.	N.D.	N.D.
D11	3.014 × 10^5^	1.161 × 10^−4^	0.385	1.257 × 10^5^	1.969 × 10^−4^	1.567

N.D. not determined.

## Data Availability

The original contributions presented in the study are included in the article/Appendix A. Further inquiries can be directed to the corresponding author.

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
