# Peer review of "Engineered Multivalent Nanobodies Efficiently Neutralize SARS-CoV-2 Omicron Subvariants BA.1, BA.4/5, XBB.1 and BQ.1.1"

_vaccines, 2024, doi:10.3390/vaccines12040417_

Round 1

Reviewer 1 Report

Comments and Suggestions for Authors

Wang et al. constructed a high-diversity nanobody phage display library and identified 9 nanobodies with high affinities to SARS-CoV-2 RBD. From the 9 nanobodies, 4 (A31, B13, B18, C2) had the ability to cross-neutralize WT, BA.1, BA.4/5 and one (A14) showed the ability to cross-neutralize additional BQ.1.1 and XBB.1. Four of the 5 nanobodies (without B18) were further selected to construct homo-, hetero-dimer or trimer nanobodies. The majority of the multimeric nanobodies in particular B13-B13-B13 showed significantly improved cross-neutralization ability against all the VOCs tested. Further studies found B13-C2, B13-B13, and B13-B13-B13 could tolerate point mutations of RBD in BQ.1.1 and XBB.1, providing evidence to support the observed broad cross-neutralization ability. Lastly, the multivalent nanobodies were found to show good thermostability, supporting drug development. This manuscript demonstrated that monomeric nanobodies not directly targeting conserved epitopes of RBD could still be used to generate multimeric nanobodies with broader cross-neutralization ability. This manuscript is well written. Data are presented in a clear and easy-to-understand way. Overall, the work is significant and supports the development of multimeric nanobodies to cross-neutralize VOCs.

Author Response

Comments and Suggestions for Authors: Wang et al. constructed a high-diversity nanobody phage display library and identified 9 nanobodies with high affinities to SARS-CoV-2 RBD. From the 9 nanobodies, 4 (A31, B13, B18, C2) had the ability to cross-neutralize WT, BA.1, BA.4/5 and one (A14) showed the ability to cross-neutralize additional BQ.1.1 and XBB.1. Four of the 5 nanobodies (without B18) were further selected to construct homo-, hetero-dimer or trimer nanobodies. The majority of the multimeric nanobodies in particular B13-B13-B13 showed significantly improved cross-neutralization ability against all the VOCs tested. Further studies found B13-C2, B13-B13, and B13-B13-B13 could tolerate point mutations of RBD in BQ.1.1 and XBB.1, providing evidence to support the observed broad cross-neutralization ability. Lastly, the multivalent nanobodies were found to show good thermostability, supporting drug development. This manuscript demonstrated that monomeric nanobodies not directly targeting conserved epitopes of RBD could still be used to generate multimeric nanobodies with broader cross-neutralization ability. This manuscript is well written. Data are presented in a clear and easy-to-understand way. Overall, the work is significant and supports the development of multimeric nanobodies to cross-neutralize VOCs.

Response: Thank you very much for taking the time to review our manuscript. We truly appreciate your accurate summary and positive evaluation of our work.

Reviewer 2 Report

Comments and Suggestions for Authors

In the submitted manuscript “Engineered Multivalent Nanobodies Efficiently Neutralize 2 SARS-CoV-2 Omicron Subvariants BA.1, BA.4/5, XBB.1 3 and BQ.1.1” nine nanobodies specific to the SARS-CoV-2 receptor-binding domain were identified, five of them were selected to their cross-neutralization ability against the SARS-CoV-2 wild-type 19 strain and the Omicron subvariants BA.1 and BA.4/5, and further engineered combining two or three nanobodies in a multivalent molecule to enhance the therapeutic potential. Results confirmed the improved affinity with spike protein, that could enhance tolerance toward escape mutations.

The topic is relevant, the manuscript is well written and clearly present the problem, the abstract and the introduction are exhaustive, and clearly reported the aim of the study. The methods are well described and valid but not very well written; the obtained results are scientifically valid, impacting but not well written.

I think that some minor revisions can improve the quality of the manuscript and after these revisions the manuscript can be accepted for publication in Vaccines. There are only only minor concern that must be considered:

- In the whole manuscript, several and several abbreviations are reported, and they make difficult the text reading and understanding, please reduce them, it would facilitate the understanding of the text for researchers not expert on this field.

Author Response

Comments and Suggestions for Authors: 

Comment 1: In the submitted manuscript “Engineered Multivalent Nanobodies Efficiently Neutralize  SARS-CoV-2 Omicron Subvariants BA.1, BA.4/5, XBB.1 and BQ.1.1” nine nanobodies specific to the SARS-CoV-2 receptor-binding domain were identified, five of them were selected to their cross-neutralization ability against the SARS-CoV-2 wild-type strain and the Omicron subvariants BA.1 and BA.4/5, and further engineered combining two or three nanobodies in a multivalent molecule to enhance the therapeutic potential. Results confirmed the improved affinity with spike protein, that could enhance tolerance toward escape mutations.

Response 1: Thank you very much for taking the time to review our manuscript. We truly appreciate your accurate summary of our work.

Comment 2: The topic is relevant, the manuscript is well written and clearly present the problem, the abstract and the introduction are exhaustive, and clearly reported the aim of the study. The methods are well described and valid but not very well written; the obtained results are scientifically valid, impacting but not well written.

Response 2: Thank you for pointing out these deficiencies. We have kindly requested an extension of the revision time from the editor in order to carefully rewrite both the methods and results sections. However, the editor informed us that they will provide professional copyediting for our manuscript if it is accepted. Additionally, we have highlighted the revisions in red in the revised manuscript.

Comment 3: I think that some minor revisions can improve the quality of the manuscript and after these revisions the manuscript can be accepted for publication in Vaccines. There are only only minor concern that must be considered:

- In the whole manuscript, several and several abbreviations are reported, and they make difficult the text reading and understanding, please reduce them, it would facilitate the understanding of the text for researchers not expert on this field.

Response 3: We are extremely grateful to you for pointing out this problem. Based on your suggestion, we have reduced the use of abbreviations such as “VHH”, “WT”, “S”, and “SPR”. Additionally, we have highlighted these modifications in red in the revised manuscript.

Reviewer 3 Report

Comments and Suggestions for Authors

This article reports on the design and evaluation of new Multivalent Nanobodies against SARS-Cov-2 variants.

The abstract contains less than 250 words and is attractive to capture the attention of readers. The GM IC50 value (20.83) of the best nanobody stands out. Perhaps a comparison with other literature values would be desirable.

The article is quite well written. The introduction is clear and fluid and the central objective of the work is highlighted.

The authors provide an adequate methodological description.

The images have been made with care. In general they are of good quality and self-explanatory.

The authors have provided the original Wester-Blot images (images 2E and 5B). They also provide supplementary material on the Construction of a VHH phage display library and the SPR binding kinetics of nanobodies to the RBD proteins of SARS-CoV-2 WT and Omicron BA.1.

I believe that the work will be of interest to readers on this important topic.

Author Response

Comments and Suggestions for Authors: This article reports on the design and evaluation of new Multivalent Nanobodies against SARS-Cov-2 variants. The abstract contains less than 250 words and is attractive to capture the attention of readers. The GM IC50 value (20.83) of the best nanobody stands out. Perhaps a comparison with other literature values would be desirable. The article is quite well written. The introduction is clear and fluid and the central objective of the work is highlighted. The authors provide an adequate methodological description. The images have been made with care. In general they are of good quality and self-explanatory. The authors have provided the original Wester-Blot images (images 2E and 5B). They also provide supplementary material on the Construction of a VHH phage display library and the SPR binding kinetics of nanobodies to the RBD proteins of SARS-CoV-2 WT and Omicron BA.1. I believe that the work will be of interest to readers on this important topic.

Response: Thank you very much for taking the time to review our manuscript. We truly appreciate your positive evaluation of our work. We fully endorse your suggestion to compare the GM IC50 value (20.83) of the best nanobody with other literature values. However, it is regrettable that there are currently no available literature reports on the GM IC50 value of a nanobody against SARS-CoV-2 WT, BA.1, BA.4/5, XBB.1 and BQ.1.1.